

# Assimilation of SCATSAR Soil Wetness Index in SURFEX 8.0 to improve weather forecasts

Stefan Schneider[1], Bernhard Bauer-Marschallinger[2]

[1]ZAMG, Vienna, 1190, Austria
[2]TU Wien, Vienna, 1040, Austria

*Correspondence to*: Stefan Schneider (stefan.schneider@zamg.ac.at)

**Abstract.** To date, in numerical weather prediction models it has only been possible to assimilate surface soil moisture data. Due to the structure of this surface soil layer in soil models, the effect of the assimilation vanishes fast. This results in a small impact on model performance. Here we present a combination of two new developments to overcome this problem. On the one hand, a new satellite based soil moisture data set is assimilated that combines the advantages of two different sensors (MetOp ASCAT and Sentinel-1 SAR) and the so-called T-value approach to estimate the soil moisture content of deeper soil layers. On the other hand, an advanced version of the well-established SURFEX soil model data assimilation software is used to ingest this new data source for improved short-range weather forecasts. Comparisons of the two data sets from satellite and soil model indicate that the estimation of deep soil moisture from superficial measurements produces reasonable estimates down to 0.5m. Assimilation experiments with a simplified Extended Kalman Filter for a model domain covering Austria shows the benefit of this new combination with improved verification scores for temperature and relative humidity forecasts at 2m above ground.

## 1 Introduction

Assimilation of surface soil moisture (SSM) in numerical weather prediction (NWP) models has proven to be useful for global models, especially in the Tropics (Dharssi et al., 2011). For mid-latitudes in global models and limited area models (LAM) in general, results are not that clear (e.g. Mahfouf, 2010; Dharssi et al., 2011; deRosnay et al., 2012; Schneider et al., 2014). The data to be assimilated can be provided by satellites or by in-situ stations. While in-situ measurements can provide profile soil moisture and they are invaluable for calibrating and validating land surface models and satellite-based soil moisture retrievals, the number of meteorological networks and stations measuring soil moisture, in particular on a continuous basis, is still limited and the data they provide lack standardization of technique and protocol (Dorigo at al., 2011). This makes it difficult to use them in data assimilation. Soil moisture measurements from satellite sensors provide



global information of SSM, but no direct information about the profile soil moisture (Wagner et al., 1999), limiting data assimilation to the superficial soil layer so far. This restrictions comes with the drawback that the changes due to data assimilation are not long-lasting in the model, as the modelled superficial soil layer is usually thin (0.01m in SURFEX (Masson et al., 2013) and TERRA_ML (Doms et al., 2011), 0.07m in H-TESSEL (Balsamo et al., 2009), 0.1m in JULES

(Best et al., 2011) and Noah-MP (Niu et al., 2011)), thus having no long-term memory. To overcome this problem, it has to be approached from two sides: On the one hand, reliable measurement data have to be provided for deeper soil layers (see chapter 2), and on the other hand, the model has to be modified to be able to assimilate soil moisture in several, deeper, soil layers (see chapter 3).

The current data assimilation investigation is the evolvement of a development that started several years ago with Mahfouf

(2010). SSM from MetOp ASCAT was assimilated in the ISBA 2-layer force-restore soil scheme (Noilhan and Mahfouf, 1996) with a simplified Extended Kalman Filter (sEKF; Mahfouf et al., 2009) with the aim to improve the ALADIN (Bubnova et al., 1995) NWP model. Draper et al. (2009) introduced another satellite data (AMSR-E) and compared the EKF and sEKF approach. Schneider et al. (2014) tested the impact on short-range precipitation forecasts in ALADIN when assimilating ASCAT SSM in the ISBA 2-layer version with the sEKF. The next development step was the 3-layer Force-

restore soil scheme, which was used to assimilate SSM and leaf area index (LAI) (Barbu et al., 2011; Fairbairn et al., 2016). The ISBA diffusion scheme (Decharme et al, 2011) with 14 soil layers finally was tested with SSM data (Parrens et al., 2014) as well as SSM and LAI data (Alberghel et al., 2017). The latter one used soil moisture in deeper soil layers as control variables for the first time.

Comparable recent studies using other forecasting models include Santanello et al. (2016), assimilating AMSR-E SSM in

WRF with en Ensemble Kalman Filter (EnKF). Like in other studies, there was found no clear, overall improvement, but a potential for future application.

The testing environment for the new approach with more observational input data is described in chapter 4, while chapter 5 contains the outcome and validation of the case studies. Chapter 6 is finishing this paper with some conclusion remarks and an outlook.

**2 Satellite data**

Within the last two decades, the knowledge on the retrieval of soil moisture content from satellite has been rapidly increasing, making these products an important observation data source for different applications, e.g. in rainfall estimation, flood risk modelling, irrigation management, and in numerical weather prediction (Dorigo and de Jeu, 2016). Nevertheless,

there are still user requirements that cannot be fulfilled with a single sensor system. One of the main issues of remotely sensed observations is the spatio-temporal coverage of the satellite sensors. There are products with a good temporal




coverage, but a poor spatial resolution (e.g. MetOp ASCAT), and vice versa (e.g. Sentinel-1 SAR), missing either detail in spatial or temporal dynamics. In order to close this scale gap, a fusion of complementing satellite products allow to create a product which combines the advantages of two measurement systems, enabling the analysis of hydrological soil processes at the kilometer scale, as convectional rainfall events. TU Wien is developing such a product, employing a data fusion

algorithm (S-1 SSM, Bauer-Marschallinger et al., in review 2) that ingests SSM data from MetOp ASCAT (Naeimi et al., 2007) and Sentinel-1 CSAR to compute the so-called Scatterometer-Synthetic Aperture Radar Soil Water Index (SCATSAR-SWI). This product (Figure 1 maps the product, its inputs, as well as soil porosity to highlight the importance of the improved resolution) infers from a joined SSM database the 1km soil profile wetness (the SWI, Wagner et al., 1999) in the topmost soil layers, down to approximately 1m. A simple model for the infiltration of the soil water to the deeper layers is

used, described by the so-called T-value as a function of the infiltration time. The SCATSWI-SWI algorithm is introduced in detail in the study of Bauer-Marschallinger et al. (in review 1), who also examined thoroughly the product's performance over the Italian region of Umbria, comparing the satellite data against in-situ and a high-resolution hydrological model. The study also assessed the dataset's ability to estimate rainfall, as its rate of change could be successfully linked to observed rainfall over Italy.

The SCATSAR-SWI dataset used for this work comes with a grid spacing of 1km and a temporal resolution of 1 day, providing data for each pixel in the grid at 12UTC, and covers an area in central Europe boxed between 0°E to 18.5°E and 43°N to 54°N, for the period January 2015 to June 2016. As input SSM datasets, the S-1 SSM product version C0101, and the H-SAF1 H110 Metop ASCAT Data Record 2016 SSM time series 12.5 km sampling was used (Bauer-Marschallinger et al., 2016).

**3 Model**

The SURFEX (SURFace EXternalisée; Masson et al., 2013) modeling system is a stand-alone soil model used in several applications in mainly meteorology and hydrology. One of the features of this model is the included data assimilation

---

[1] http://hsaf.meteoam.it/soil-moisture.php

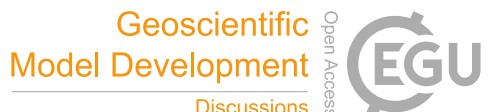

package SODA (Surface Offline Data Assimilation) that can be used to update the prognostic soil variables with measurements of temperature and relative humidity 2m above ground (T2M, RH2M), LAI, SSM and snow water equivalent (SWE). The assimilation method used for this purpose is the sEKF (Mahfouf, 2010). To be able to assimilate SWI data in several modelled soil layers, the most advanced version of the soil model, the ISBA diffusion scheme (Decharme et al.,

2011), has to be used. By default, this soil scheme comes with 14 soil layers. Until now, the assimilation software has been just used to work with observations for SSM and the control variables WG1 and WG2 (soil moisture content in SURFEX layers 1 and 2). SODA inside SURFEX 8.0 has been modified both to be able to read measurement data for the soil layers 2 to 6 and to run the sEKF for the prognostic variables WG3 to WG6.

To test the impact of the data assimilation on atmospheric forecasts, the soil model has to be coupled to a NWP model. For

this purpose, the convection-permitting LAM AROME (Seity et al., 2011) has been chosen. AROME (version CY40T1) is used operationally at ZAMG, thus the operational model setup was used for the data assimilation experiments. SURFEX (version 7.3) is coupled to AROME as soil model in this setup. To use the ISBA diffusion scheme in the NWP model, some code modifications were necessary.

The forecast and data assimilation processing chain consist of several steps: First, an atmospheric forecast has to be run. For

the tests in chapter 4, this is a 24-hourly forecast, starting at 12UTC every day. This forecast provides the atmospheric forcing (temperature, relative humidity, wind speed and direction, precipitation, long- and short-wave radiation and $CO_2$) for SURFEX. This forcing is used to run several stand-alone soil model forecasts for the same time range as the AROME forecast mentioned above: one unperturbed reference forecast and one (or more), slightly perturbed, run(s). The number of runs is depending on the number of control variables which are affected by data assimilation. These soil forecasts are the

input for the sEKF which is computing the new analysis for the end of the forecasting period. This updated soil analysis is the new initial state of the soil, used both for the next AROME and the next SURFEX soil forecast, and so on.

To run the assimilation experiments, a model domain covering Austria has been set up (see Figure 2). It consists of 259x133 grid points. The horizontal grid spacing is 2.5km and in the vertical, the model has 90 levels. This resolution is the same as in the operational setup of ZAMG and has proven to run stable. Austria was chosen due to the very dense network of





meteorological stations for verification purposes, whereas this choice comes with the drawback of very complex topography

in large parts of the forecasting domain, challenging both the model and the satellite data.

**4 Comparison of SCATSAR-SWI and SURFEX soil moisture fields**

As Sentinel-1 is in operational mode only since 2015, the processed SCATSAR-SWI dataset available for the studies

contains just 1.5 years of data (January 2015 to June 2016). This leads to the problem that the time series which are

necessary for the bias correction of the satellite data are relatively short, containing just one year of data (January to

December 2015). For the CDF matching approach (Reichle and Koster, 2004), which was chosen for the bias correction, this

is a rather poor data set which unfortunately does not allow for distinguishing seasonal effects in the bias. To apply the CDF-

matching, several working steps are necessary to make satellite and model data comparable.

SURFEX soil moisture data are available with a temporal granularity of 1h. To match them to the SCATSAR-SWI data,

which are availability once per day, the SURFEX data are temporally averaged for 24h, with 12UTC being the reference

time. This averaging is done for each soil layer separately.

In the second step, the quality flags coming with the satellite SCATSAR-SWI data are used to filter out low quality

measurements. The following thresholds have been defined: SSF = 1 (indicates that the grid box is free of snow/ice),

QFLAG >= 160 (160 of this quality flags means that the data quality is 80%) and CMASK (rho=0.3; a mask from the soil

moisture data fusion, applicable for e.g. cities, lakes, wood, and mountains) >= 150. The remaining high-quality SCATSAR-

SWI satellite data are interpolated to the model grid by building the mean of the 4 satellite grid cells (1x1km) being located

closest to the model cell (2.5x2.5km) center. With this approach it is ensured that the spatial dimension of measurements and

modelled data is of a comparable range.

SCATSAR-SWI data are provided as relative values, ranging from 0 (dry minimum) to 100 (wet maximum). In the last pre-

processing step, they are rescaled to water [m³] / soil [m³], corresponding to domain in SURFEX. For this computation, the

soil properties (sand and clay content) of SURFEX are used. SCATSAR-SWI values of 100% (0%) are set equal to the

saturation point (wilting point) of the model grid point. All SCATSAR-SWI values in between are scaled linearly

accordingly. To set the lower limit of the SCATSAR-SWI to the wilting point is reasonable for deep soil layers, as the





usually do not dry out completely. For the uppermost model layer with a vertical extent of just 1cm, it would be also an option (not tested here) to set the lower limit to 0, as this soil layer can totally dry out. This would change the range of the soil moisture data slightly, but the impact on the CDF matching is negligible.

The result of these processing steps is a set of time series of modelled and measured soil water content for each model grid
point and each model layer. These time series are used to compute the 4th-order regression equations for the CDF-matching, whereas grid points with less than 100 data pairs (modelled soil moisture – measured SWI) have been dismissed. This is mainly the case for mountainous regions in the western part of Austria.

So far, there has been no detailed comparison of the SCATSAR-SWI with a soil model on such high resolution so it was not clear a priori if the suggested representative soil depths for the SCATSAR-SWI are reasonable for larger regions. To find the
best agreement between model and satellite data, the correlation coefficient (CC) for each combination of SCATSAR-SWI and SURFEX WG has been computed for 2015. Table 1 shows the mean positive CC for the whole domain, meaning that the average for all model grid points with positive CC has been computed. Positive CC is mainly found over flatlands. This comparison shows an overall good agreement between SCATSAR-SWI (index T) and SURFEX (index WG). For T001 and T005, the correlation is highest with WG1. For T010, T015 and T020 the correlation is highest with WG4 and WG5. T040
and WG5 as well as T060 and WG6 have the best correlation. Just for T100, it is not WG8 that gives the best correlation, but WG6. From these findings it can be assumed that the T-value approach is reasonable down to about 0.5m soil depth with the rule of thumb of a penetration speed of one centimeter per day. Deeper down in the soil, this relation is not applicable anymore. There is no in-situ soil moisture measurement network available in Austria, thus both data sets are not compared to in situ measurements. So it cannot be stated here if one of the two data sets or both of them causes this discrepancy.
Based on this result, it has been decided to use the following SCATSAR-SWI data for the corresponding SURFEX soil moisture content: T001 - WG1, T005 – WG2, T010 - WG3, T020 - WG4, T020 – WG5 and T060 - WG6. T015 is omitted for the assimilation as there is no corresponding soil layer. T100 is omitted as it shows the same characteristic as T060.

**5 Data assimilation case studies**

Forecasting convection is one of the key research topics in LAM modelling, so the late spring/early summer of 2016 has been defined as testing period for the assimilation experiments. This period was chosen for two reasons. First, it is a part of the convective season in Austria which lasts from May to September (Schulz et al., 2005). Second, good data coverage for SCATSAR-SWI data is given for this period, while for the first quarter of 2016, many grid points are lacking valid data due

to snow cover or frozen soil conditions in Austria. Taking into account the fact that the data assimilation cycle needs to have a spin-up phase. May to June 2016 has been defined as the validation period for the data assimilation experiments, while April 2016 was used as spin-up phase.

There are three assimilation experiments besides the reference run without data assimilation (EXP1):

EXP2 uses T001 as observation input data and WG1 as control variable to be modified due to the data assimilation.

EXP3 also uses T001 as observation, but this time, WG1 to WG6 are used as control variables.

EXP4 uses several satellite data (T001, T005, T010, T020, T040 and T060) and WG1 to WG6 as control variables.

EXP2 is the reproduction of the well-investigated superficial soil moisture assimilation and is the reference data assimilation experiment to be improved. The comparison of EXP2 and EXP3 is used to find out if there is an additional value of the fine vertical resolution (14 instead of 2 vertical soil layers) of the ISBA diffusion scheme when providing just superficial soil

moisture as observation data source. EXP4 finally is the experiment to test the expected additional benefit of the SCATSAR-SWI data set on model performance.

For quantification of the impact of the data assimilation, a comprehensive in-situ soil moisture measurement network is necessary, but however not existing over Austria. To overcome this, atmospheric screen-level parameters are used instead for validation. This is argued by the fact that soil moisture has a significant impact on the distribution of sensible and latent

heat flux and evaporation near the ground, thus influencing temperature and relative humidity at 2m above ground. Both variables are well captured by ZAMGs SYNOP station network. In addition, precipitation is chosen as validation variable, as soil moisture is supposed to influence convection which is often causing precipitation. As mentioned above, there is one forecast run per day, starting at 12UTC with a forecast range of 24 hours and hourly output variables for temperature, humidity and precipitation. In the following, the average for these 60 short-range forecast runs (May-June 2016) is presented

and discussed.



As mentioned, SCATSAR-SWI data for assimilation are mainly available over lowlands due to the measurements constraints and the quality control. Thus the 265 stations which provide measurements for the verification (see Figure 2) have been separated in several height classes: up to 300m (46 stations), 300-500m (59), 500-700m (51), 700-900m (34) and 900m and above (75). Statistical values (bias and root mean squared error (RMSE)) are computed from hourly station measurements

and model output fields.

For precipitation, no significant positive impact due to data assimilation can be detected on average for the testing period. This finding is in good agreement with Schneider et al. (2014). The RMSE, computed for all 265 stations in Austria, is 1.05 for all four experiments. EXP4 is the worst for stations below 300m with an RMSE of 1.04, compared to 1.01 for the reference run EXP1, while it is the best experiment for stations between 300 and 500m (1.16 for EXP4 vs. 1.18 for EXP1).

None of these differences are statistically significant which has been tested with the Mann-Whitney-test (Mann and Whitney, 1947), as implemented in Python's SciPy. RMSE values are directly correlated with the average precipitation amount produced by the different experiments. EXP4 produces approximately 5% more precipitation than EXP1-3 for stations below 300m, but 1% less for stations between 300 and 500m. Above this height, there is no difference between the four experiments, neither in RMSE nor in the amount of precipitation. This is in good agreement with the fact that there is just a

limited number of SCATSAR-SWI values for higher elevations to be assimilated, so there should be no strong modifications of the soil fields over mountainous regions. EXP2 and EXP3 show no clear difference to EXP1 with regard to RMSE. This is also true for the bias, which is slightly negative for all experiments but without clear differences between the assimilation experiments and the reference run.

When investigating single forecast runs, it can be stated that convective precipitation patterns over lowlands are modified by

the assimilation, but obviously, these modifications do not systematically improve the forecast quality. The increase in precipitation should in principle improve the forecast quality, as the model tends to underestimates precipitation amounts in lowlands by about 25%. Unfortunately, the modifications in the initiation and lifecycle of the cells are, at least for the tested period, randomly distributed.

For the 2m variables temperature (T2M) and relative humidity (RH2M), forecast quality for EXP1 has a clear diurnal cycle.

There is a pronounced negative bias for T2M during daytime (12-19UTC and 04-12UTC) which means that the air near the



ground is too cold in the model during day while it is slightly too warm during night (see Figure 3 (right), dark blue line). In combination with this, there is strong positive bias for RH2M detected during daytime in the reference model run (see Figure 4 (right), dark blue line).

According to the basic energy balance at the surface, the downwards incoming radiant energy is partitioned into upwards

sensible and latent heat flux and a small downward ground heat flux. If a lot of water is available, the latent heat flux is growing at the expense of sensible heat flux which results in a colder and wetter atmospheric ground layer (Sellers et al., 1997). So the results of EXP1 are clearly indicating that the model soil during the testing period is on average too wet.

While this problem of the model is not solved by EXP2 and EXP3 (see Figure 3 and Figure 4, red and light blue lines), it is significantly reduced in EXP4 (green lines in Figure 3 and Figure 4). The bias is smaller both for T2M and RH2M during

daytime, and this reduction also causes an improved RMSE for both variables. During night time, RMSE is reduced for RH2M, but not for T2M. As there is a clear change in the statistical measures in the morning and evening, it is obvious that the model has a problem in modelling the correct diurnal temperature and humidity trend. The positive effect of reduced evaporation during daytime is kept for RH2M, so both statistical measures are improved. For T2M, stronger heating of the atmospheric ground layer has no positive impact during night which might be correlated with this problem of the forecasting

system to model the diurnal temperature cycle correctly.

The investigation of screen level parameters indicate that the assimilation of SCATSAR-SWI improves forecast quality and it is also obvious from comparing EXP3 and EXP4 that data from several depths are necessary to achieve this improvement. To determine the effect of the assimilation on the water content of the model soil, the analysis increments have been summarized for the whole investigation period of 60 days for each grid point and each soil layer where WG has been used as

control variable. On average, these increments are negative, which means that water is removed from the soil. This is in perfect agreement with the initial finding that the reference run (EXP1) is too moist on average. Comparing the increments for different experiments show that the drying effect in EXP4 is by far the strongest one. In the lowlands of eastern Austria, approximately 0.01kg/kg water is removed per daily run from the uppermost layer in EXP4 (Figure 5, right), which is a factor of ~$10^3$ more than in EXP3 (Figure 5, left). A similar behaviour can be seen for all layers down to WG6, the deepest



soil layer where SCATSAR-SWI data have been assimilated (Figure 6). Here the factor is about $10^2$, which means that the drying of the soil is again much more efficient in EXP4.

## 6 Conclusions and outlook

Soil moisture is a crucial variable in meteorological forecast models. To benefit from the globally available remotely sensed soil moisture data sets, operational assimilation of these data is state-of-the-art in several met services around the globe. So far, this assimilation was restricted to the superficial soil layer due to the lack of measurement data for deeper soil layers. This leads to a weak impact of the assimilation as the memory of this superficial layer is naturally very short. The new SCATSAR-SWI data set which includes the T-value approach is the first step to overcome this limitation.

In the present study, the SCATSAR-SWI data set has been compared to a nation-wide soil model. The both independent data sets show a linear relationship down to a depth of about 50cm which is clearly indicating that the simple concept of the infiltration depth is a good estimation. For the deepest layer T100, the correlation is still highest for the SURFEX layers of about 50cm depth. From the investigation here it is not clear if this discrepancy for the deep soil arises from the measurement data set, the SURFEX model or a combination of both. One source of error might be the vertical constant soil

(sand and clay content) profiles that are used both in the model and for the T-value approach, whereas the latter even uses no grain size distribution at all. This will have to be investigated in more detail with the aid of in-situ measurements or an additional soil model.

The findings of this comparison have been used to define the setup of the data assimilation experiments. As a consequence, the sEKF assimilation software in SURFEX was adapted for the additional observation data in the deeper model soil layers 2

to 6. The assimilation experiments show the clear benefit of the additional information provided by the SCAT-SWI data set. The wet bias in the reference run can be reduced most efficient by an assimilation configuration where measurements for several layers are taken into account as shown by the comparison for screen level parameters (T2M and RH2M) in Austria. These findings indicate that the assimilation of SCATSAR-SWI in several depths is beneficial to improve model performance compared to the state-of-the-art approach of assimilating just SSM. For precipitation, no clear impact could be



found on average. For single events, the precipitation patterns are influenced by the assimilation, but there is no systematic improvement. So the fact that soil moisture is necessary but not sufficient information for the initiation of convection in the NWP model will force us to further investigate this problem.

One possible approach would be the use of both SCATSAR-SWI and some temperature information for assimilation or at

least use model ground temperature (TG) as additional control variable. This should lead to a better balance between TG and WG and maybe a more reasonable partitioning between sensible and latent heat fluxes after the assimilation step of the forecast cycle.

The use of the SCATSAR-SWI in operational weather forecast models requires that the generation of this data set is operationalized. The Copernicus Global Land Service (CGLS2) of the European Commission disseminates operationally and

freely soil moisture products together with other bio-geophysical variables to enable the monitoring of the global vegetation, water and energy budget. Acknowledging the utility of the SCAT-SAR-SWI, the CGLS is currently preparing the operational product dissemination of this data set, featuring coverage in an initial phase over Europe, and subsequently globally.

**Code availability**

The SURFEX model code is accessible and can be downloaded on open source (http://www.cnrm-game-meteo.fr/surfex/, last access: 31 October 2018). This platform is regularly updated; however, the model developments described in the paper have yet to be taken into account in the latest SURFEX version (v8.1). For all further information or access to real-time code modifications, please follow the procedure in order to open the SVN account provided via the previous link. The routines modified with respect to the sEKF assimilation are available upon request submitted to the corresponding author.

**Data availability**

Model experiment data are available upon request submitted to the corresponding author.

---

[2] https://land.copernicus.eu/global/





**Author contribution**

Stefan Schneider designed and carried out the NWP experiments, made the validation and wrote the related chapters.

Bernhard Bauer-Marschallinger is responsible for the scientific work concerning the creation of the SCATSAR-SWI data

and wrote the related chapter.

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

|  |  |  | SURFEX soil moisture |  |  |  |  |  |  |  |  |
|---|---|---|---|---|---|---|---|---|---|---|---|
|  |  |  | WG1 | WG2 | WG3 |  | WG4 | WG5 | WG6 | WG7 | WG8 |
|  |  | Depth [cm] | 0-1 | 1-4 | 4-10 |  | 10-20 | 20-40 | 40-60 | 60-80 | 80-100 |
| SCAT SAR-SWI | T001 | ~1 | 0.45 | 0.43 | 0.41 |  | 0.38 | 0.31 | 0.25 | 0.20 | 0.17 |
|  | T005 | ~5 | 0.46 | 0.44 | 0.45 |  | 0.44 | 0.41 | 0.34 | 0.28 | 0.23 |
|  | T010 | ~10 | 0.45 | 0.44 | 0.45 |  | 0.46 | 0.44 | 0.38 | 0.32 | 0.27 |
|  | T015 | ~15 | 0.46 | 0.44 | 0.45 |  | 0.47 | 0.47 | 0.42 | 0.36 | 0.30 |

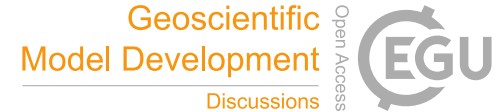



| | | | | | | | | | |
|---|---|---|---|---|---|---|---|---|---|
| T020 | ~20 | 0.48 | 0.46 | 0.47 | | 0.49 | 0.49 | 0.45 | 0.39 | 0.33 |
| T040 | ~40 | 0.52 | 0.50 | 0.51 | | 0.55 | 0.58 | 0.54 | 0.49 | 0.42 |
| T060 | ~50 | 0.50 | 0.49 | 0.51 | | 0.55 | 0.61 | 0.59 | 0.54 | 0.48 |
| | | | | | | | | | | |
| T100 | ~100 | 0.47 | 0.45 | 0.46 | | 0.51 | 0.60 | 0.61 | 0.55 | 0.48 |

**Table 1: Mean positive correlation coefficient between different SCATSAR-SWI and SURFEX WG data sets.**

(a) 25 km ASCAT SSM | Morning Coverage
MetOp-A ASCAT | 2015 09 01 | 08:14 & 09:53

(b) 1 km Sentinel-1 SSM | Full Day Coverage
Sentinel-1A | 2015 09 01 | 05:17

(c) 1 km SCATSAR-SWI | T=5 | Daily Coverage
Sentinel-1A & MetOp-A ASCAT | 2015 09 01 | 12:00

(d) Soil Parameter | Example: Porosity 0-5 cm
ISRIC SoilGrids

Soil Moisture [%]    0   25   50   75   100

No Data
Water Bodies

Porosity [%]    37   39.5   42   44.5   47



**Figure 1: Example images for SCATSAR-SWI and its inputs, and soil porosity for the Austrian area. (a) Metop ASCAT SSM data at 25km resolution from two overpasses on the morning of 2015-09-01, where brown colours indicate dry surface conditions, and blue wet soils. White areas are not covered by the sensor. (b) Similarly, SSM from Sentinel-1 at 1km resolution on the same day. (c) Similarly, the SCATSAR-SWI data at 1km resolution on the same day. (d) For comparison, the porosity of the soil layer at 0-5cm depth from ISRIC SoilGrids (Hengl et al., 2017), illustrating that the SCATSAR-SWI and Sentinel-1 SSM are much closer to the scale of soil characteristics than the ASCAT SSM.**

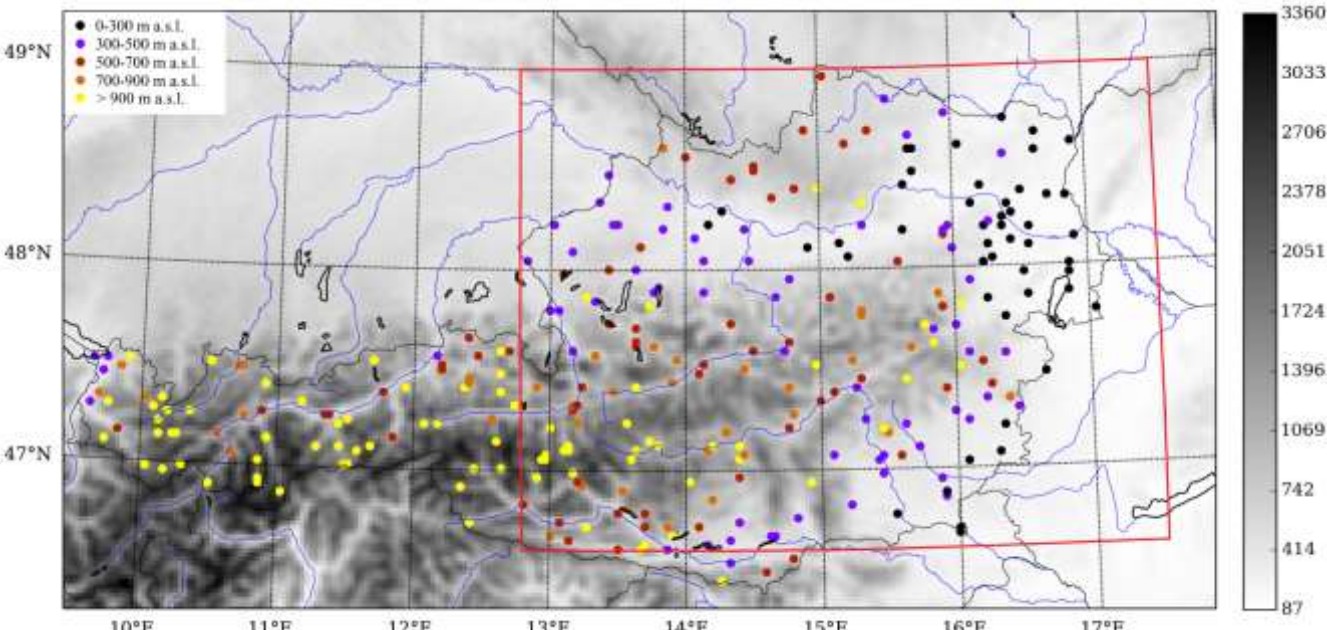

**Figure 2: Model domain, covering mainly Austria, with SURFEX model topography in meters. The dots indicate the location of the stations used for verification. The red rectangle indicates the subdomain used in Figure 5 and Figure 6.**





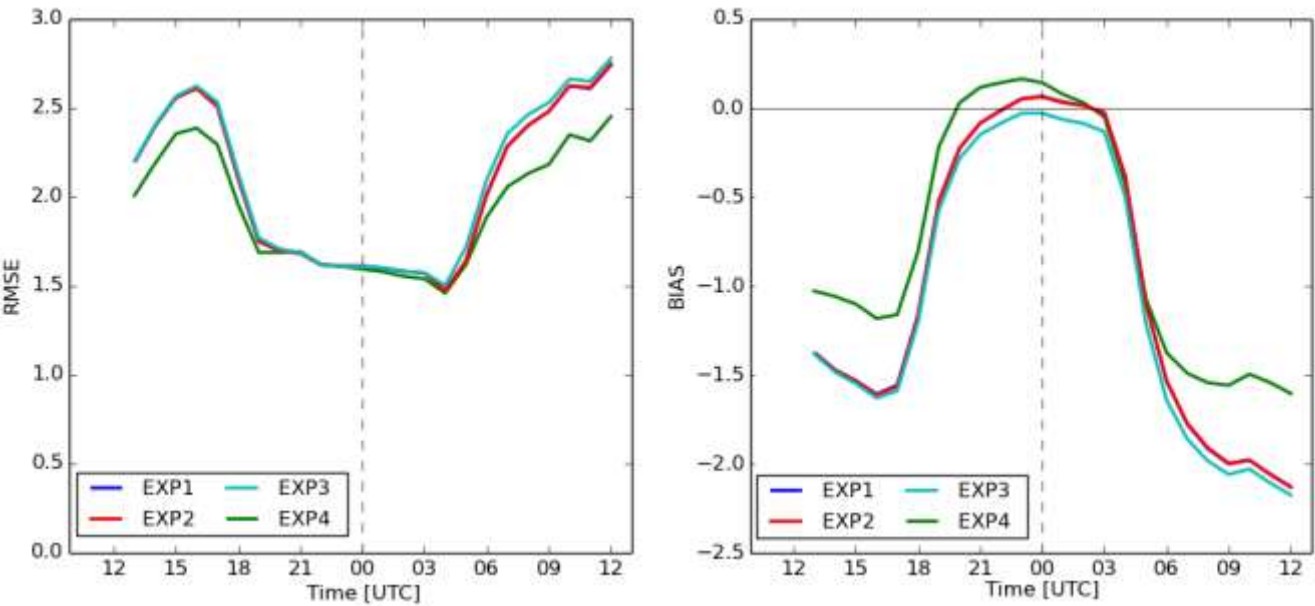

**Figure 3: RMSE (left) and Bias (right) for T2M for forecasts from 20160501-20160629, averaged for all stations below 300m.**

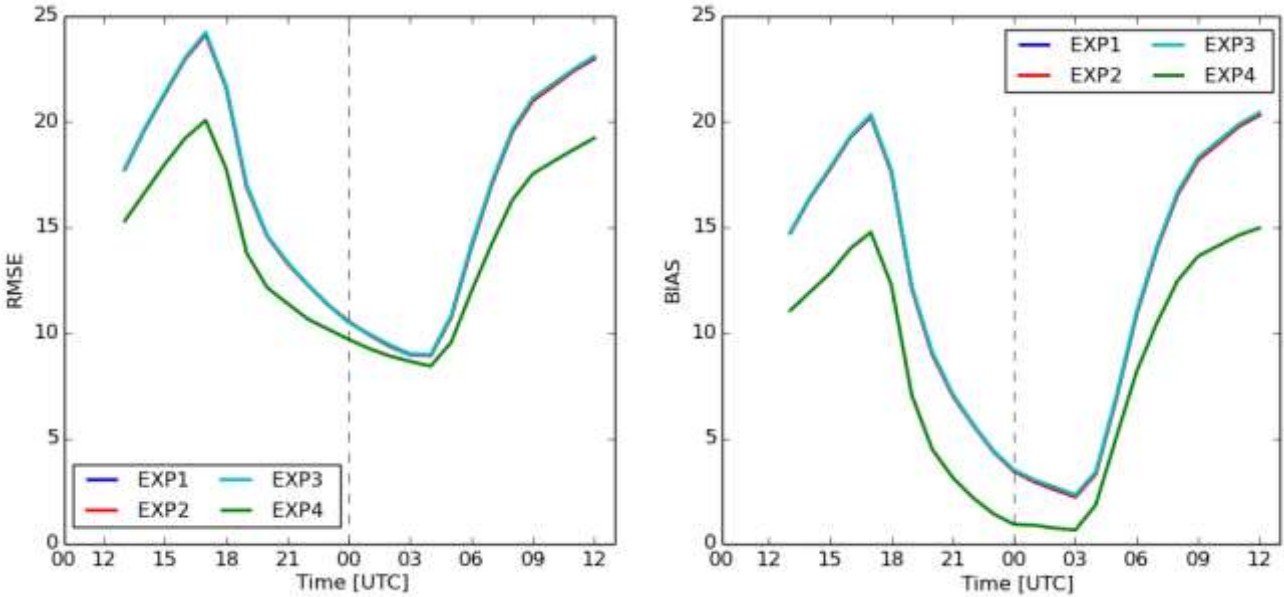

**Figure 4: Like Figure 3, but for relative humidity 2m above ground. EXP1, EXP2 and EXP3 are literally equal most of the time, so**
5 **the lines are overlapping.**



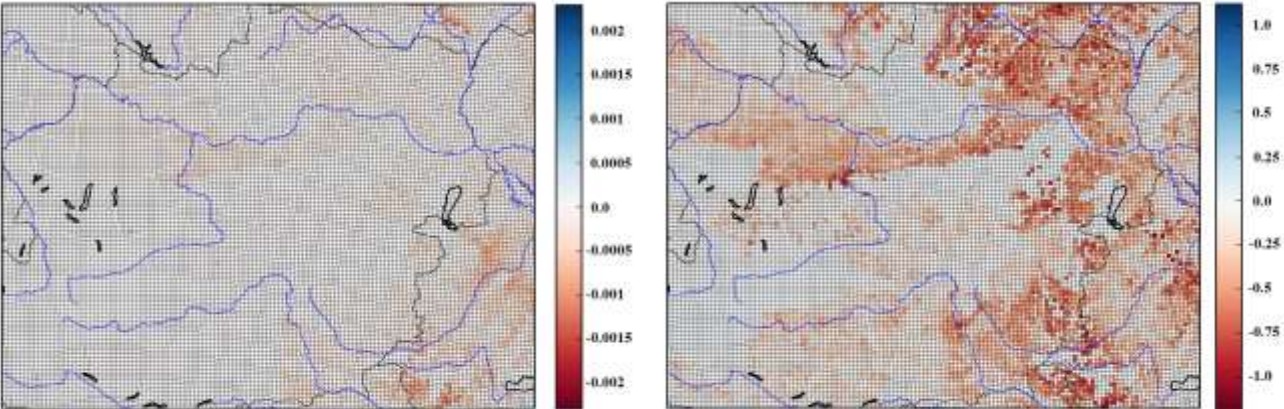

**Figure 5: Summarized analysis increments (60 analyses from May-June 2016) for WG1 in EXP3 (left). Drying of the soil due to data assimilation is about a factor of $10^3$ weaker than for EXP4 (right).**

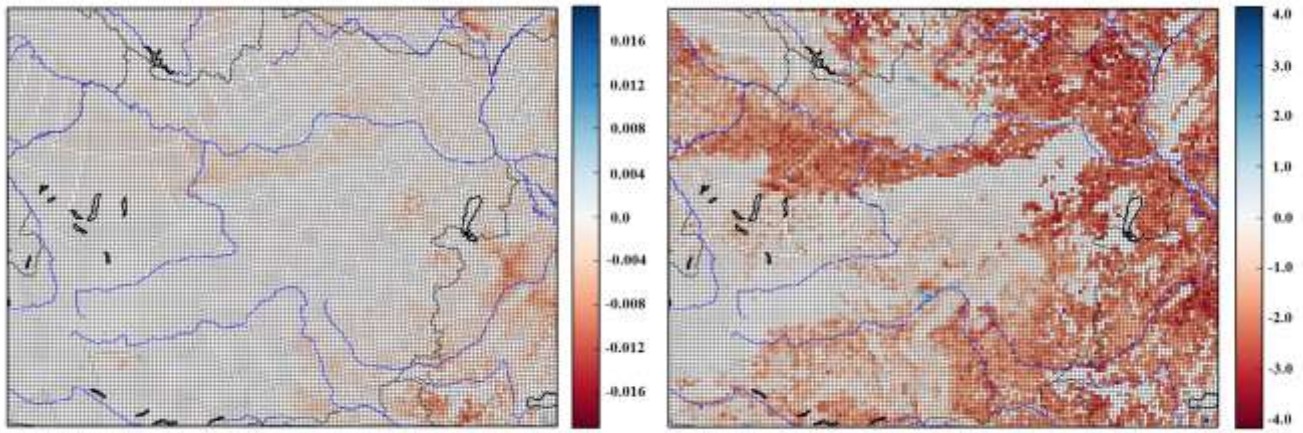

5  **Figure 6: Like Figure 5, but for WG6. For this layer, a factor 100 is between the increments, so EXP4 is extremely more efficient in removing the wet bias in the model soil.**