# Peer review of "Assimilation of SCATSAR Soil Wetness Index in SURFEX 8.0 to improve weather forecasts"

_Geoscientific Model Development, 2018_

## Referee Comment (RC1) · Anonymous Referee #1 · 24 Jan 2019

Comments by Reviewer

on

Manuscript https://doi.org/10.5194/gmd-2018-273

Assimilation of SCATSAR Soil Wetness Index in SURFEX 8.0 to improve weather forecasts

by

Stefan Schneider, Bernhard Bauer-Marschallinger

General comments: ———————————————— This Paper is concerned with the assimilation of a satellite based based soil moisture product in a numerical weather prediction modelling system containing a SURFEX surface model using the ISBA 14 layer diffusion scheme. The satellite-based soil moisture product is based on two different kind of sensors, on-board different satellites and a Kalmanfiler-based data surface data-assimilation technique is used. Results point at benefit fo NWP forecast quality from using the satellite product in surface data assimilation.

The Paper points at an important point, which is how to initialize the deeper soil layers in the model. The general idea of the study is very interesting but unfortunately, in my opinion, the investigation is not extensive enough and suffers from some limitations. In addition the Paper is too short and technical. I therefore recommend that the Paper is subject to major modifications. After that these modifications have been carried out I consider the Paper ready for a another evaluation, prior to future potential publication. My main general concerns regarding the Paper are:

1. The Paper in is current form is too short. In particular a more detailed description on the procedure for merging the satellite data from different satellites and instruments and the procedure for that is needed. Now there is only a short description with a reference to Papers in review without providing doi or link. I think more information in the Paper itself is needed.

2. In several places I consider the description in the Paper too technical. For example you do not need to provide the surfex version in the title. Furthermore, in the title the acronym SCATSAR does not need to be mentioned,. It can be replaced with Scatterometer-Synthetic Aperture Radar or you reformulate the title. Other examples when I consider text to technical 'as implemented in Python's SciPy' and 'One possible approach would be the use of both SCATSAR-SWI and some temperature information for assimilation or at least use model ground temperature (TG) as additional control variable.' I think no need to mention 'as implemented in Python's SciPy' and that grounf temperature is called 'TG0' in the model used.

3. I think some more details of sEKF and its limiations and similarities as compared with other assimialtion methods suchs as optimal interpolation and EnKF should be provided. Is it for example appropriate with a linear assumption to use a series of 24 h perturbed to calculated the Jacobians? I think not. I think the dependence of Jacobians and thus results on length of perturbed forecast when calculating Jacobians shoud be caaried out and presented. Perhaps possible to apply 24 h cycle but calculating Jacobians for shorter period, please elaborate and extend experiments I suggest. Are other assimilaltion methods like EnKF more appropriate and is a long spin-up period needed also for data assimilation? If Jacobians where larger when calculating based on for example 3 h forecats then experiment 3 would perhaps be more competative with experiment 4? Then Jacobians in combination with information from top soil moisture layer only would be sufficient possibly, or not? Very interesting question and you are really trying to answer in paper but please consider the importance of Jacobian calculation.

4. Please add some more Figures. I suggest an overview Figure of the data assimilation procedure. It is good to have mean accumulated soil moisture increments as in Figures 5 and 6. However to look at magnitudes also perhaps absoulute values of increments would be useful? In addition mean and absolute Jacobian or gain matrix values should be added. I propose as well a number of smaller modifications of already existing Figures.

More specific comments: ————————-

1. Page 1, line 1-2. I suuggest removing version number and acronym SCATSAR

2. Page 1, lines 4-5 I think abreviations ZAMG and TU should be instead written out both here and then first time mentioned in text.

3. line 15, and many cases like this. Change '0.5m' to '0.5 m'

4. Page 3, lines 1-2 Can you please provide some more details on why the temporal

resolution is higher for METOP/ASCAT as compared to resolution of Sentinel-1 SAR? Perhaps provide some more details on spatial resolution and footprint as well and discussion of difference as compared to model horizontal resolution?

5. Page 3, lines 9-11 Please provide link to report on methodology and include substantial part of it in this paper to strengthen the Paper.

6. Page 4., lines 15-25 Please provide a lot of more details on assimilation methodology. What is meant by 'slightly' perturbed? I think an overview schematic Figure would be good.

7. Page 7, lines 5-15 Please make clear what is meant by 'spin-up phase' in the sence of how model is configured. I think the 4 experiments could more clear be presented in bullet form.

8. Page 12 The two reports in review by Bauer-Marschallinger is not cleary accesable yet. Please provide link and include relevant part in this paper as well.

9. Page 18, Figures 3-4 Please provide units on vertical axes (K and %) as well as in Figure texts. In addition I think it would be preferable to calculate and present standard deviation rather than rmse.

10. Page 18, Figures 5-6 Please provide units in Figure texts.

---

## Referee Comment (RC2) · Anonymous Referee #2 · 20 Mar 2019

This study is very interesting and promising. However, the paper does really not do it justice. I feel like it was written quickly and that the authors skimmed over some key explanations, presentation of the results and proper discussions. Additionally I disagree with several statement made by the Authors (my points are discussed below). I believe that there are too many things to do to reach the potential of this paper. I recommend the editor to reconsider the submission this paper. That is why my recommendation is reject and encourage to resubmit. Please find below an attempt to help.

Major issues

P.1, L.7: 'to date [. . .] soil moisture data', it is a strong statement, moreover I don't

see why and rather disagree. The T-value approach exists from nearly two decades (Wagner et al., 1999, see also Wagner's PhD Thesis)...My understanding is that it acts as a low pass filter, smoothing the values from the surface as well as adding a small time shift (?). As you mention at latter stage you match the climatology of the estimated (rather than observed) soil moisture to the one of the model. By doing it you want to keep the short term variability of the assimilated product. This short term variability is probably lost in T-value approach when using to high T-values (but you may want to prove me wrong). P.6, L.16, I also believe you should justify more the use of the T-values themselves, giving a general rule on how to translate a given T-value to a certain soil depth is currently not possible since this depends strongly on the application and the soil composition of the area of interest. This is not discussed enough in the manuscript. I suggest that the T-values methodology needs to be developed as the reader needs more explanation to fully understand why choices were made and what are the implications of those choices. P.6, L.16, to understand the link between model and observations, could you please show the estimated Jacobian and interpret the results, I believe it is interesting. More information on the DA set up and methodology is needed, I am not sure that I understood which was the model equivalent of the observations. Data assimilation description is rather poor (what about the model and observation errors, this a key aspect of DA), a diagram showing the general flow and design of the DA will be very helpful as well. Please add an experimental set up section. P.7, first paragraph reads like one of the rational of the study if not the rational, this story about forecasting convection belongs to the introduction (?) P.10, 'due to lack of measurements', I disagree, as stated above the T-value approach exists for long time and your sentence implies that you have a brand new observation type which is wrong. Reference section: several references mentioned in the text are not in this section (the opposite is also true). General comment on figures: what is GMD policy for figures? I suppose you need scale and North on all of them (?)

Other issues

P.1, L.21, 'results are not that clear', please rephrase. P.2, L.3, 'usually thin (0.01m in SURFEX' OK however many recently published paper using SURFEX land surface data assimilation do not consider this superficial layer and use the second layer of soil of SURFEX (or rather ISBA, the land surface model in SURFEX) as model equivalent of the surface soil moisture. P.2, L.9, 'years ago with Mahfouf (2010)', please rephrase P.2, L.18, You should double check as I am wondering if your approach has been tested by Barbu et al., 2011 or 2014. P.3, L.4, what is TU WIEN? P3., L.7, where does the soil porosity comes from? P.3, L.10, explain T-value better (see comment above), what is SCATSWI-SWI ? P.4, L.5, I always got confused with soil layering, better to have a diagram to clearly show the detailed layering scheme. P.4., L.11, what is ZAMG? P.4, L.12, I am confused, title says SURFEX 8.0, here it is written version 7.3 and acknowledgement section refers to 8.1, please clarify. P.4, L.13, which modifications? P.5, L.10, what are those working steps? P.5, L.11-13, you refer more to your experimental set up that to SURFEX in general (?) A section describing the experimental set up is always welcomed. P.5, L.15, what is SSF? Acronyms are required, see GMD publication policy. P.5, L.16, what is CMASK? P.5, L.17, what is 150? P.5, L.21-22, using which soil layer? P.6, L.8-14, This paragraph needs more discussions. P.6, L.21, see my comment above on soil layering. P.7, better to describe the experiments in a Table. P.8, you must explain/discuss those results. P.9, L.17, use of 'obvious' seems too strong to me in this context. P.10, L.7, what is 'met' P.10, L.10-14, you have to go further and look for example to the seasonal scale and possible decoupling between surface and deeper layers. Table 1, add information on the sampling, I personally find CC values rather low (?) could you comment on that? SURFEX soil moisture has been evaluated in many papers.

---

## Author Comment (AC1) · 16 Apr 2019

Answers to Review 1

General comments: ——————————— This Paper is concerned with the assimilation of a satellite based based soil moisture product in a numerical weather prediction modelling system containing a SURFEX surface model using the ISBA 14 layerdiffusionscheme. Thesatellite-basedsoilmoistureproductisbasedontwodifferent kind of sensors, on-board different satellites and a Kalmanfiler-based data surface data-assimilation technique is used. Results point at benefit fo NWP forecast quality from using the satellite product in surface data assimilation.

The Paper points at an important point, which is how to initialize the deeper soil layers in the model. The general idea of the study is very interesting but unfortunately, in my opinion, the investigation is not extensive enough and suffers from some limitations. In addition the Paper is too short and technical. I therefore recommend that the Paper is subject to major modifications. After that these modifications have been carried out I consider the Paper ready for a another evaluation, prior to future potential publication. My main general concerns regarding the Paper are:

1. The Paper in is current form is too short. In particular a more detailed description on the procedure for merging the satellite data from different satellites and instruments and the procedure for that is needed. Now there is only a short description with a reference to Papers in review without providing doi or link. I think more information in the Paper itself is needed.

*By now, both papers on the Sentinel-1 soil moisture input, and on the SCATSAR-SWI merging method, have been published. They describe in detail the motivation, the instruments, and the procedures, complemented by evaluation experiments.*

2. In several places I consider the description in the Paper too technical. For example you do not need to provide the surfex version in the title.

*This is a request from the editor as the paper is planned to be a part of the SURFEX special issue of GMD.*

Furthermore, in the title the acronym SCATSAR does not need to be mentioned,. It can be replaced with Scatterometer-Synthetic Aperture Radar or you reformulate the title.

*The abbreviation SCATSAR is now explained in the abstract.*

Other examples when I consider text to technical 'as implemented in Python's SciPy' and 'One possible approach would be the use of both SCATSAR-SWI and some temperature information for assimilation or at least use model ground temperature (TG) as additional control variable.' I think no need to mention 'as implemented in Python's SciPy' and that grounf temperature is called 'TG0' in the model used.

*As a part of the SURFEX special issue of GMD, we suppose that using the SURFEX variable names in the text is useful for other scientists working with SURFEX thus we would like to keep this in the text. Mentioning the software that was used should make the work better reproducible in our opinion.*

3. I think some more details of sEKF and its limiations and similarities as compared with other assimialtion methods suchs as optimal interpolation and EnKF should be provided.

*We will add some literature regarding this topic in the introduction section.*

Is it for example appropriate with a linear assumption to use a series of 24 h perturbed to calculated the Jacobians? I think not. I think the dependence of Jacobians and thus results on length of perturbed forecast when calculating Jacobians shoud be caaried out and presented. Perhaps possible to apply 24 h cycle but calculating Jacobians for shorter period, please elaborate and extend experiments I suggest.

*It is true that the experimental set-up of this investigation could be enlarged to include other, not yet sufficiently answered, questions with regard to the Extended Kalman Filter approach. The length of the time window to compute the Jacobians is one of them. The focus of this paper is on the impact of the assimilation of soil moisture on NWP forecast performance, so several scientific questions with regard to the assimilation method have been omitted.*

*The linear assumption has to be questioned in principle in the assimilation in a weather model as it is well-known that NWP models usually have a diurnal cycle of forecast quality. Atmospheric events like precipitation are clearly have non-linear impacts on error characteristics. Nevertheless, in practice the assumption leads to reasonable results so it is used in this investigation.*

Are other assimilaltion methods like EnKF more appropriate and

*Indeed this is a very interesting question. Answering this question by comparing assimilation experiments with sEKF and EnKF would be a stand-alone scientific publication and is therefore not planned to be included in this paper.*

is a long spin-up period needed also for data assimilation?

*A spin-up is necessary no matter if data assimilation is applied or not. As we have no reliable measured information on the vertical temperature and soil moisture profiles in the model domain, the model has to be initialised with an educated guess which comes in our case from the operational NWP system at ZAMG. This model comes with 2(3) layers for soil temperature (moisture) and needs some vertical interpolation. Due to the long memory of the deep soil layers the model needs several days/weeks to transfer these interpolated values to the diffusion model physic. This is explained now in more detail in the description of the experimental setup.*

If Jacobians where larger when calculating based on for example 3 h forecats then experiment 3 would perhaps be more competative with experiment 4? Then Jacobians in combination with information from top soil moisture layer only would be sufficient possibly, or not? Very interesting question and you are really trying to answer in paper but please consider the importance of Jacobian calculation.

*Jacobians will be investigated and the findings will be presented in the revised paper.*

4. Please add some more Figures. I suggest an overview Figure of the data assimilation procedure.

*We will add a flow chart on the data assimilation procedure.*

It is good to have mean accumulated soil moisture increments as in Figures 5 and 6. However to look at magnitudes also perhaps absolute values of increments would be useful?

*Figures 5 and 6 are showing the accumulated increments for all 60 days of the investigation period. As the increments during this time period are negative almost everywhere, the accumulated increments and absolute values of increments are more or less the same except the algebraic sign.*

*We will try to reformulate the text to avoid confusions between mean and accumulated increments for the readers.*

In addition mean and absolute Jacobian or gain matrix values should be added.

*We will investigate the Jacobians and add our findings to the paper. This is currently still work under progress so we cannot give a detailed answer on the outcomes right now.*

I propose as well a number of smaller modifications of already existing Figures.

*The modifications mentioned below in the specific comments have been included in the paper.*

More specific comments: ———————

1. Page 1, line 1-2. I suuggest removing version number and acronym SCATSAR

*The version number is a request by the editor, thus we will keep it in the title.*

2. Page 1, lines 4-5 I think abreviations ZAMG and TU should be instead written out both here and then first time mentioned in text.

*This has been changed in the paper.*

3. line 15, and many cases like this. Change '0.5m' to '0.5 m'

*This has been changed throughout the paper*

4. Page 3, lines 1-2 Can you please provide some more details on why the temporal resolution is higher for METOP/ASCAT as compared to resolution of Sentinel-1 SAR? Perhaps provide some more details on spatial resolution and footprint as well and discussion of difference as compared to model horizontal resolution?

*We summarized now the key characteristics of the two sensors in the text.*

5. Page 3, lines 9-11 Please provide link to report on methodology and include substantial part of it in this paper to strengthen the Paper.

*The SWI-approach is defined and discussed in Wagner et al, 1999. We streamlined the text for more clarity.*

6. Page 4., lines 15-25 Please provide a lot of more details on assimilation methodology. What is meant by 'slightly' perturbed? I think an overview schematic Figure would be good.

*This section has been modified in the paper. For the perturbations, numerical values (0.001 kg/kg) are mentioned now in the text.*

7. Page 7, lines 5-15 Please make clear what is meant by 'spin-up phase' in the sence of how model is configured. I think the 4 experiments could more clear be presented in bullet form.

*The model soil (ISBA-DIF with 14 soil layers) is initialised with values from the soil of the operational forecast model of ZAMG. This is AROME with the ISBA force-restore soil with 2(3) soil layers for temperature (soil moisture). Values from these layers are interpolated to the 14 soil layers, leading to unrealistic vertical gradients. With the spin-up phase it is possible to bring the soil temperature and moisture back to physically consistent values.*

*The experiment set up is now presented in tabular form in the paper*

8. Page 12 The two reports in review by Bauer-Marschallinger is not cleary accesable yet. Please provide link and include relevant part in this paper as well.

*This has been included in the paper.*

9. Page 18, Figures 3-4 Please provide units on vertical axes (K and %) as well as in Figure texts. In addition I think it would be preferable to calculate and present standard deviation rather than rmse.

*Units are added in the paper.*

*RMSE is generally used to measure the error of prediction so we suppose that it is a good statistical measure to investigate NWP model forecast performance.*

10. Page 18, Figures 5-6 Please provide units in Figure texts.

*This has been added in the captions.*

Answers to Review 2

This study is very interesting and promising. However, the paper does really not do it justice. I feel like it was written quickly and that the authors skimmed over some key explanations, presentation of the results and proper discussions. Additionally I disagree with several statement made by the Authors (my points are discussed below). I believe that there are too many things to do to reach the potential of this paper. I recommend the editor to reconsider the submission this paper. That is why my recommendation is reject and encourage to resubmit. Please find below an attempt to help.

Major issues

> P.1, L.7: 'to date [...] soil moisture data', it is a strong statement, moreover I don't see why and rather disagree. The T-value approach exists from nearly two decades (Wagner et al., 1999, see also Wagner's PhD Thesis)...My understanding is that it acts as a low pass filter, smoothing the values from the surface as well as adding a small time shift (?). As you mention at latter stage you match the climatology of the estimated (rather than observed) soil moisture to the one of the model. By doing it you want to keep the short term variability of the assimilated product. This short term variability is probably lost in T-value approach when using to high T-values (but you may want to prove me wrong).

*Indeed, the SWI-approach uses a temporal filter that aims for modelling the infiltration of the SSM content into deeper soil layers - it estimates the water content of soil profile layer from the local SSM history. The top surface layer is in direct contact with the atmosphere and its water content is thus temporally highly dynamic. Naturally, the temporal SM dynamics are decreasing with depth. This is modelled by the T-value parameter in the SWI-modelling, acting as a local and soil-dependent measure for the infiltration time. With this, short-term variability in the surface layer (but not shorter than daily) is integrated in the SWI signal, visible in the top layers, but more and more "smoothed" in deeper layers, reflecting the assumptions on the soil infiltration.With the bias correction via CDF matching we want to keep as much of the original signal as possible, like the seasonal cycle, the anomaly signals and also of course the short-term variability, but removing the different mean values. Of course, the T-value approach leads to a smoothing of the variable, but "short-term" has to be related to the time scales of the soil layer of interest, so there is also short-term variability in deeper layers.*

> P.6, L.16, I also believe you should justify more the use of the T-values themselves, giving a general rule on how to translate a given T-value to a certain soil depth is currently not possible since this depends strongly on the application and the soil composition of the area of interest. This is not discussed enough in the manuscript. I suggest that the T-values methodology needs to be developed as the reader needs more explanation to fully understand why choices were made and what are the implications of those choices.

*We agree that there is no general rule how the T-values relate to soil depth, as this depends on the soil type and layering. There are attempts to understand this relationship (e.g. Paulik et al., 2014) but they can be understood only as recommendation - it is left up to the user to decide what to use locally. We emphasize this more in the text.*

*The analysis on page 6, with results in Table 1, examines the relationship between T-values and soil depth for the Austrian area and the SURFEX soil layers, as we statistically determines the best-matching pairs of T-value and soil layer.*

P.6, L.16, to understand the link between model and observations, could you please show the estimated Jacobian and interpret the results, I believe it is interesting. More information on the DA set up and methodology is needed, I am not sure that I understood which was the model equivalent of the observations. Data assimilation description is rather poor (what about the model and observation errors, this a key aspect of DA), a diagram showing the general flow and design of the DA will be very helpful as well. Please add an experimental set up section.

*We will investigate the Jacobians and add our findings to the paper. This is currently still work under progress so we cannot give a detailed answer on the outcomes right now.The section describing the assimilation set-up will be reformulated to make the details more clearly for the reader and we will add an flow chart on the data assimilation procedure. The model and observation error standard deviation are set to 0.2, this is now added to the text.*

P.7, first paragraph reads like one of the rational of the study if not the rational, this story about forecasting convection belongs to the introduction (?)

*This is mentioned here as an argument why an early summer period is chosen as testing period as this time of year is typically characterised by many convective events in the testing region Austria.*

P.10, 'due to lack of measurements', I disagree, as stated above the T-value approach exists for long time and your sentence implies that you have a brand new observation type which is wrong.

*This has been rephrased in the paper*

Reference section: several references mentioned in the text are not in this section (the opposite is also true).

*Before resubmitting the paper, all references will be checked again.*

General comment on figures: what is GMD policy for figures? I suppose you need scale and North on all of them (?)

*We are not aware of this but we will discuss this topic with the editor.*

Other issues

P.1, L.21, 'results are not that clear', please rephrase.

*This has been rephrased in the paper*

P.2, L.3, 'usually thin (0.01m in SURFEX' OK however many recently published paper using SURFEX land surface data assimilation do not consider this superficial layer and use the second layer of soil of SURFEX (or rather ISBA, the land surface model in SURFEX) as model equivalent of the surface soil moisture.

*You are right but due to the fact that the second soil layer in the ISBA diffusion scheme has a depth of 3 cm (from 1-4 centimetres below the surface), the statement about the short memory is also true for the case where this second layer is used. ISBA has been added in the text.*

P.2, L.9, 'years ago with Mahfouf (2010)', please rephrase

*This has been rephrased in the paper*

P.2,L.18,You should double check as I am wondering if your approach has been tested by Barbu et al., 2011 or 2014.

*You are right that there are two papers of Barbu et al. which are both dealing with this topic. The second one from 2014 has been added now in the paper.*

P.3, L.4, what is TU WIEN?

*Technische Universität Wien. The abbreviation is explained now in the text.*

P3., L.7, where does the soil porosity comes from?

*It is described in the caption of the figure, but we added it now also in the text.*

P.3, L.10, explain T-value better (see comment above), what is SCATSWI-SWI ?

*Good points. We now give much more attention in the text to the SWI approach, explaining the temporal filtering.*

*We fixed the typo "SCATSWI-SWI".*

P.4, L.5, I always got confused with soil layering, better to have a diagram to clearly show the detailed layering scheme.

*To avoid the duplication of an already existing figure, the reference to the original one in Decharme et al., 2013 has been added in the paper.*

P.4., L.11, what is ZAMG?

*Zentralanstalt für Meteorologie und Geodynamik. The abbreviation is explained now in the text.*

P.4, L.12, I am confused, title says SURFEX 8.0, here it is written version 7.3 and acknowledgement section refers to 8.1, please clarify.

*For the assimilation of soil moisture data, SURFEX v8.0 (with some code modifications described in the paper) is used in stand-alone ("offline") mode. The resulting improved analyses are used to initialise the atmospheric model AROME to compute the atmospheric forecasts that are validated in the paper. AROME uses SURFEX as soil model and in the AROME-version operationally available at ZAMG (CY40T1), SURFEX v7.3 is included. So different processing steps of the assimilation experiments are computed with different SURFEX versions but the configuration settings are the same in both versions wherever possible. SURFEX v8.1 mentioned in the "Code availability" section is referring to the most current release of SURFEX. This version has not been used in the experiments.*

*This has been rephrased in the paper (section 3 and Code availability section)*

P.4, L.13, which modifications?

*The code documentation of the HIRLAM consortium is added as reference to explain the necessary modifications to run the ISBA-DIF scheme in CY40T1.*

P.5, L.10, what are those working steps?

*The working steps are described starting from P.5 L11. The text has been rephrased so the three working steps are easier to detect.*

P.5, L.11-13, you refer more to your experimental set up that to SURFEX in general (?) A section describing the experimental set up is always welcomed.

*Yes, this is describing the experimental set up. This has been rephrased in the paper.*

P.5, L.15, what is SSF? Acronyms are required, see GMD publication policy.

*Surface State Flag. It describes the state of the surface with respect to frozen soil conditions, or snow cover. We add a description accordingly.*

P.5, L.16, what is CMASK?

*The SCATSAR-SWI's correlation mask. It flags pixels where the correlation between SCAT and SAR input is low and thus not eligible for the SWI data fusion.*

P.5, L.17, what is 150?

*This means that grid points with cities, lakes, wood and mountains are masked out.*

P.5, L.21-22, using which soil layer?

*The soil properties of SURFEX are vertically constant, so the same values for clay and sand are applied to all SCATSAR-SWI T-values.*

P.6, L.8-14, This paragraph needs more discussions.

*The discussion of this topic (similarities between the two data sets) is ongoing until L18. We would need more details what is missing from the reviewers point of view.*

P.6, L.21, see my comment above on soil layering.

*A reference to the figure showing to soil layering is provided now in the paper.*

P.7, better to describe the experiments in a Table.

*The experiment set up is now presented in tabular form in the paper*

P.8, you must explain/discuss those results.

*P8 is mainly referring to precipitation results which show no statistically significant differences between the 4 experiments. We add a paragraph on the theory of precipitation – soil moisture feedback mechanisms which led us to the assumption that a better representation of soil moisture could lead to better convective precipitation forecasts.*

P.9, L.17, use of 'obvious' seems too strong to me in this context.

*The results of EXP3 and EXP4 are statistically significant different, so we think that "obvious" is justifiable in this context.*

P.10, L.7, what is 'met'

*This should be "meteorological". It has been changed in the paper.*

P.10, L.10-14, you have to go further and look for example to the seasonal scale and possible decoupling between surface and deeper layers. Table 1, add information on the sampling, I personally find CC values rather low (?) could you comment on that? SURFEX soil moisture has been evaluated in many papers.

*We agree that the seasonal scale would be of interest, but with a data set that contains just values for one and a half year (Jan 2015 – June 2016) the basic population is too small to produce results that have statistic relevance. So we omitted this investigation.*

*The spatial sampling of the data in Table 1 is 2.5 km, this has been added in the caption.*

*CC values presented in the table are the average values over all grid points in the domain where there is a positive correlation. For single grid points, CCs up to 0.98 are available.*

---

## Editor Comment (EC1) · Jeffrey Neal (Editor) · 26 Apr 2019

Dear Stefan,

Thank you for your response to the reviewer's comments. Please accept my apologies for taking a few days to review your response over the Easter break.

Having reviewed the comments and your response to the reviewers I'm in agreement with Reviewer 2 that the paper has potential, however the presentation of the research is a long way from the required standard and will therefore need to be treated as if a new submission if a revised version is submitted. This is partly because of the exten-

sive nature of the revisions required, but also due to the need for additional analysis to address some of the reviewer's questions, which appears to be ongoing. If a significantly improved revised manuscript is submitted, I will be seeking additional referees for further review in addition to those who have commented already.

Further to the reviewer's comments, as a non-expert I find the second half of the introduction difficult to follow and overly focused on a set of technical changes without really explaining the motivation or outcome of these developments. Furthermore, the end of the introduction doesn't explain the motivation or aims of the paper in a clear way as these are distributed in the text between the introduction and chapter 2. The structure of the introduction therefore needs major revision.

The version number in the title is indeed a requirement of the journal as you state. I would agree with reviewer 2 that the maps in Figure 1 should include a scale and north arrow.

The expert reviewers have provided excellent feedback and the latter technical sections of the paper and I encourage you to address all reviewer comments

Best wishes, Jeff